# COVID-19 Vaccine Acceptance during Pregnancy and Influencing Factors in South Korea

**DOI:** 10.3390/jcm11195733

**Published:** 2022-09-28

**Authors:** Heesu Yoon, Bo Yun Choi, Won Joon Seong, Geum Joon Cho, Sunghun Na, Young Mi Jung, Ji Hye Jo, Hyun Sun Ko, Joong Shin Park

**Affiliations:** 1Department of Obstetrics and Gynecology, Seoul St. Mary’s Hospital, College of Medicine, The Catholic University of Korea, Seoul 06591, Korea; 2Department of Obstetrics and Gynecology, School of Medicine, Kyungpook National University, Daegu 41404, Korea; 3Department of Obstetrics and Gynecology, Korea University College of Medicine, Seoul 08308, Korea; 4Department of Obstetrics and Gynecology, Kangwon National University Hospital, School of Medicine, Kangwon National University, Chuncheon 24289, Korea; 5Department of Obstetrics and Gynecology, Seoul National University College of Medicine, Seoul 03080, Korea; 6Department of Obstetrics and Gynecology, Yeouido St. Mary’s Hospital, College of Medicine, The Catholic University of Korea, Seoul 07345, Korea

**Keywords:** COVID-19, pregnancy, vaccination, acceptance

## Abstract

Pregnant women were excluded from vaccination against Coronavirus 2019 (COVID-19) until September 2021 in South Korea. Although vaccination for pregnant women started in October 2021, vaccine acceptance in pregnant women is yet unknown. This cross-sectional study aimed to investigate COVID-19 vaccine acceptance during pregnancy and influencing factors. An anonymous survey was distributed in obstetrics departments to all pregnant or postpartum women, during the prenatal or postpartum visit. The proportion of self-reported COVID-19 vaccination during pregnancy among 436 women was 26.6%. Pregnancy-related independent factors influencing maternal COVID-19 vaccination were “received vaccine information about from obstetrics and gynecology (OBGYN) doctors” (OR 3.41, 95% CI 2.05–5.65), “cohabitant COVID-19 vaccination” (OR 2.43, 95% CI 1.06–5.59), and “second trimester” (OR 7.35, 95% CI 1.54–35.15). In women who did not want to get vaccinated, the most common reason for COVID-19 vaccination hesitancy was concern that COVID-19 vaccine might affect the fetus (91.7%, 243/266), followed by distrust in COVID-19 vaccine effectiveness (42.6%, 113/266). This study showed that providing information about maternal COVID-19 vaccination, especially by OBGYN doctors, is crucial for increasing vaccination coverage in pregnant women. Providing updated evidence of COVID-19 vaccine efficacy and safety in pregnant women may be also helpful for increasing vaccine acceptance.

## 1. Introduction

With the emergence of new severe acute respiratory syndrome coronavirus (SARS-CoV-2) variants of concern such as the delta and omicron variants, the coronavirus disease 2019 (COVID-19) epidemic is entering a new phase, and there have been 383,509,779 confirmed cases of COVID-19, including 5,693,824 deaths, globally, as of 3 February 2022 [1]. Although several vaccines and their booster shots have been developed and approved following the first emergency use authorization of the Pfizer-BioNTech COVID-19 vaccine in December 2020, the vaccine coverage of pregnant women is substantially lower than that of the general population [2,3].

It has been reported that pregnant women infected with SARS-CoV-2 are associated with increased risk for intensive care unit admission, mechanical ventilation, extracorporeal membrane oxygenation, and death, compared to non-pregnant women [4,5,6]. In addition, SARS-CoV-2 infection has been reported to be related to a higher rate of adverse obstetric outcomes such as preeclampsia, preterm birth, and stillbirth [7,8]. Despite the serious consequences, pregnant women were excluded from the phase 3 trial of mRNA COVID-19 vaccines [9]. Limited evidence at the time of emergency use authorization led to inconsistent vaccination guidelines during pregnancy [10].

However, post-authorization studies have demonstrated accumulating evidence of the efficacy and safety of the COVID-19 vaccine during pregnancy. Preliminary findings from three American vaccine safety monitoring systems (v-safe surveillance system, c-safe pregnancy registry, and Vaccine Adverse Event Reporting System (VAERS)) indicated no obvious safety signals regarding maternal COVID-19 vaccination [11]. Other studies also demonstrated that the risks associated with pregnancy, delivery, and newborn complications including spontaneous abortion of pregnant women with COVID-19 vaccination were comparable to those in non-vaccinated pregnant women [12,13,14,15,16,17]. Vaccinated pregnant women showed significantly lower risk of SARS-CoV-2 infection, as with the vaccinated general population [18,19,20]. In addition, some evidence suggested that maternal vaccination during the third trimester has a potential protective effect on neonates through placental immune transfer [20,21,22].

Recent guidelines from the Centers for Disease Control and Prevention (CDC), the American College of Obstetrics and Gynecologists (ACOG), and the Society for Maternal-Fetal Medicine (SMFM) strongly recommend pregnant people to receive vaccination, as well as a booster dose against COVID-19 [23,24,25]. Regardless of the changed new guidelines, pregnant women are still reluctant to get vaccinated. Though such vaccine hesitancy during pregnancy is not new, in the case of COVID-19, maternal vaccination seen difficulty due to insufficient initial evidence and serious side effect reports in the media [26]. This led to low understanding along with increasing fear of vaccination in pregnant women [27].

While the initial COVID-19 vaccination program in South Korea commenced in February 2021, pregnant women have been included in the vaccination program with BNT162b2 (Pfizer-BioNTech) or mRNA-1273 (Moderna) COVID-19 vaccines since 18 October 2021 [28]. Therefore, understanding the current perception and determinants of pregnant/postpartum women toward COVID-19 vaccination is critical for future measures or discussion of COVID-19 vaccination in pregnancy. The primary aim of this study was to characterize the acceptability and attitude toward COVID-19 vaccine uptake in pregnancy and assess the factors associated with vaccination among pregnant/postpartum women in South Korea.

## 2. Materials and Methods

### 2.1. Study Population and Recruitment

The survey was conducted from 1 January 2022 to 30 April 2022. The questionnaires for pregnant women were distributed to pregnant or postpartum women within 6 weeks after birth during routine antenatal care visits at a mix of public and private clinics or hospitals located in South Korea. Four hospitals in Seoul, one hospital in Daegu, one hospital in Kangwon, and one clinic in Jeolla were included. Participation in this survey was voluntary, and no financial or other incentives were offered. Response to the survey implied consent. Information provided by the participants was voluntary and possibly incomplete. 

### 2.2. Survey Questionnaires

The questionnaires for pregnant women were adapted from a previously self-administered questionnaire for influenza vaccination [29] composed by a multidisciplinary study team that included obstetrics and gynecology (OBGYN) doctors, biomedical statisticians, and pregnant women. A pilot survey involving pregnant women was conducted to ensure questionnaire comprehensiveness. The survey included questions about maternal demographics and socioeconomic characteristics including age, body mass index, underlying disease, pregnancy duration, parity, natural conception or use of assisted reproduction, education level, occupation, and the administrative district of residential areas. Inclusion criteria were pregnant women over six weeks’ gestation with confirmed fetal heartbeat by ultrasonography and postpartum women within six weeks after delivery. Pregnant and postpartum women were asked the following: (1) whether they received information about COVID-19 vaccination; (2) information sources; (3) whether they received COVID-19 vaccination during pregnancy including gestational period at vaccination; (4) how much they were worried about the risk of COVID-19 infection during pregnancy and risks for mothers and fetuses of COVID-19 infection; (5) who are the high-risk groups among pregnant women when they get COVID-19 infection; (6) whether they know that antibodies by vaccination can transfer to the fetus through the placenta and umbilical cord; (7) reasons for not getting vaccinated; (8) factors influencing future vaccination; (9) influenza vaccination status during pregnancy; (10) COVID-19 vaccination status of their cohabitants. General information about COVID-19 vaccination for pregnant women was sent by text message and mailing from regional public health offices, when the vaccination program for pregnant women commenced. The women were classified according to self-reported COVID-19 vaccination status during pregnancy. Response options for Questions (1), (3), (6), (9), and (10) were yes or no. Questions about information sources, high-risk groups, reasons for not getting vaccinated, and factors influencing future vaccination permitted multiple responses. Response options for questions related to (4) were provided as a 5-point Likert scale. Ethical approval for the protocol and questionnaires was granted by the Institutional Review Boards of the Catholic University of Korea (KC21QIDE0934) and Seoul National University (2203-101-1308).

### 2.3. Sample Size Calculation

The sample size for the survey of pregnant or postpartum women was calculated with the following assumptions: the proportion of women having received at least 1 shot or more of COVID-19 vaccine during pregnancy was expected to be around 20% [27], with a confidence interval of 95% and an alpha of 0.05. When we assumed the response rate as 60%, the initial calculated minimum sample size was 410 participants. Given the incomplete response rate or full vaccination rate before pregnancy (30%), however, 533 pregnant or postpartum women were recruited to meet the minimum sample size. Data collection was stopped when the minimum number of responses for the analyses was reached.

### 2.4. Data Analysis

All data analyses were performed using the R software (R Foundation for Statistical Computing, Vienna, Austria. http://www.r-project.org/ (accessed on 1 May 2022)) v.4.2.0. Continuous variables of age and pre-pregnant BMI of pregnant women are presented as the mean ± standard deviation and compared using Student’s *t*-test or Welch’s *t*-test. The data distribution was determined by the *F*-test. All other variables were categorical data, which were expressed as counts and percentages, and compared using the Chi-squared test. To assess influencing factors associated with vaccine uptake by pregnant women, we calculated the odds ratios (ORs) and 95% CIs using logistic regression analysis. Variables with a significant cutoff, *p* < 0.05, between vaccinated and unvaccinated groups in univariate analyses were included in multivariate analyses, after adjustment for maternal age, occupation, and pregnancy period. Statistical significance was set at *p* < 0.05.

## 3. Results

### 3.1. Sociodemographic Characteristics of Pregnant or Postpartum Women

A total of 436 pregnant (87.8%, 383/436) or postpartum (12.2%, 53/436) women were included in the analysis, after excluding 29 women with incomplete responses and 68 women who already got vaccinated before pregnancy. The survey completion rate was 93.8% (436/465). The questionnaires were collected online (45.0%, 196/436) or via paper survey (55.0%, 240/436). The majority were living in the capital region (63.8%), and 87.4% held a college degree or above. The geographical distribution of total respondents is presented in Appendix A. Among 436 pregnant or postpartum women, 116 women accepted COVID-19 vaccination during pregnancy, showing an acceptance rate of 26.6%. Sociodemographic characteristics between the acceptance and non-acceptance groups are presented in Table 1. There were significant differences in occupations and pregnancy period between the two groups (*p* = 0.020 and *p* = 0.001, respectively). Maternal age, residence, education level, conception method, and influenza vaccination rate during pregnancy showed no significant differences between the two groups.

### 3.2. COVID-19-Related Characteristics of Pregnant or Postpartum Women

COVID-19-related characteristics regarding COVID-19 and vaccine are presented in Table 2. Women informed about COVID-19 vaccine during pregnancy were more likely to accept vaccination (*p* = 0.001). Specifically, receiving information from an OBGYN doctor or public health office increased acceptance (*p* < 0.001 and *p* = 0.03, respectively). Those who accepted vaccination had a significantly higher rate of cohabitant vaccination and awareness of antibody transfer to the fetus (*p* = 0.008 and *p* = 0.023, respectively). Those who were worried about COVID-19 infection during pregnancy were less likely to accept COVID-19 vaccination (*p* = 0.044).

### 3.3. Influencing Factors for Maternal COVID-19 Vaccination

Results of univariate/multivariate logistic regression analysis for acceptance of the COVID-19 vaccine during pregnancy are presented in Table 3. In the univariate analyses, the variables of “medical personnel”, “second trimester”, “third trimester”, “informed about COVID-19 vaccine during pregnancy”, “received vaccine information from OBGYN doctor”, “received vaccine information from public health office”, “cohabitant got COVID-19 vaccination”, and “aware that antibodies of vaccination can transfer to fetus” significantly increased vaccine acceptance, while “worried of COVID-19 infection during pregnancy” significantly decreased vaccine acceptance (Table 3). However, in multivariate analyses, “second trimester” (OR 7.35, 95% CI 1.54–35.15), “getting vaccination information from OBGYN doctor” (OR 2.08, 95% CI 2.05–5.65), and “Cohabitant previously got COVID-19 vaccine” (OR 2.43, 95% CI 1.06–5.59) were only associated with COVID-19 vaccination acceptance.

### 3.4. Barriers against COVID-19 Vaccination and Factors Associated with Future Vaccination

Women in the non-acceptance group (*n* = 320) were asked why they did not receive vaccination, and multiple responses were allowed. There were 266 (83.1%) women who answered, “I did not want to have COVID-19 vaccination” and 2 (0.6%) women who answered, “I did not know about vaccination”. There were 76 (23.8%) women who were waiting for more pregnant women to get vaccinated, and 23 (7.2%) women responded that they had no recommendation about vaccination from a doctor. Women who did not want to get vaccinated responded to the multiple response questions about reasons for not wanting vaccination. The most common reason for COVID-19 vaccination hesitancy was concern that the COVID-19 vaccine might affect the fetus (91.4%, 243/266), followed by distrust in the COVID-19 vaccine’s effectiveness (42.5%, 113/266) and concern that the COVID-19 vaccine might affect one’s self (38.3%, 102/266) (Figure 1). Figure 2 depicts replies to the question “Whose information and recommendation would affect your future vaccination?” Among 99 respondents, 86 (86.9%) answered that information and recommendation from an OBGYN doctor would influence future COIVD-19 vaccination.

## 4. Discussion

### 4.1. Main Findings

This study indicated a lower rate of acceptance of COVID-19 vaccination among pregnant women than in the general population, which was reported as 80% in South Korea, until 29 October 2021 [29]. There were no significant differences in maternal age, education level, residence, and combined diseases. In univariate analysis, medical personnel, vaccine information, vaccine information from OBGYN doctors or a public health office, COVID-19 vaccinated family member, awareness of possible protection through transferring immunity to the baby, second or third trimester of pregnancy, and worry about COVID-19 infection during pregnancy were significant. However, we identified independent influencing factors as having received “vaccine information about from OBGYN doctors” (OR 3.41, 95% CI 2.05–5.65), living with individuals having received COVID-19 vaccination (OR 2.43, 95% CI 1.06–5.59), and being in the second trimester of pregnancy as the main predictors (OR 7.35, 95% CI 1.54–35.15), in multivariate logistic regression analysis. In women with non-acceptance of COVID-19 vaccine, 83.1% of women responded that they do not want to get vaccinated. Vaccine safety, especially anxiety about the possibility of side effects for the fetus was a primary concern. Over 40% of women with non-acceptance of COVID-19 vaccine presented distrust in the vaccine’s effectiveness. However, over 80% of women with non-acceptance of COVID-19 vaccine responded that information and recommendation from OBGYN doctors will affect their vaccinations in the future.

### 4.2. Acceptance Rates Compared to Other Maternal Vaccinations

The acceptance rate of the COVID-19 vaccine was not associated with influenza vaccination during pregnancy in this study. In recent studies, acceptance rates of influenza and Tdap during pregnancy were 63% and 67%, respectively, in South Korea [30,31]. In those studies, “getting informed” and “informed by OBGYN doctors” were significantly associated with vaccine acceptance. However, “getting informed” was not associated with COVID-19 vaccine acceptance in this study. Several reasons can be speculated such as: shorter history of COVID-19 vaccines than those of influenza and Tdap, relatively low incidence of COVID-19 infection in pregnant women in South Korea until June 2021, unexpected adverse events in young adults after the vaccination in the media, such as myocarditis, thrombosis, or other events with an unknown causal relationship, and so on [32]. In addition, the initial COVID-19 vaccination program excluded pregnant women, and the COVID-19 vaccination program for pregnant women in South Korea started later than other developed countries [28,33,34,35]. Therefore, it seems that information about COVID-19 vaccination by OBGYN doctors is especially important in pregnant women.

### 4.3. Acceptance Rates Compared to Other Pregnant Populations

In a previous multinational study including 5294 pregnant women, the COVID-19 vaccine acceptance level varied by country [36]. High acceptance rates over 60% were seen in Mexico, the Philippines, Saudi Arabia, and India; middle acceptance rates of about 45–60% were seen in Italy, Spain, the United Kingdom, and Canada; low acceptance rates below 45% were seen in the USA, Australia, Russia, and France [36,37,38,39]. A lower educational level and lower income were associated with non-acceptance of COVID-19 [27,40]. In this study, there was no significant difference in the educational levels between the acceptance and non-acceptance groups. The strongest predictors of vaccine acceptance included confidence in vaccine safety or effectiveness, and protecting their baby was the most common reason for acceptance and refusal of the COVID-19 vaccine in other pregnant populations, which were similar in the population of this study [27,39,40]. Other predictors in a survey from 16 countries were worrying about COVID-19, belief in the importance of vaccines to their own country, mask wearing compliance, trust in public health agencies, and attitudes towards routine vaccines [36]. In this study, women in the non-acceptance group were more worried about COVID-19, compared to women in the acceptance group, in the univariate analysis. However, it was not significant in the multivariate analysis. In addition, there were no significant differences whether they were informed by a public health office, the media, non-OBGYN doctors, and family members or friends in this study. Because 86.9% of women responded that information and recommendation from OBGYN doctors would affect future COVID-19 vaccination, providing information and guidelines from academic committees and public health offices to OBGYN doctors seems more important to increase vaccine acceptance in South Korea, compared to in other countries. 

### 4.4. Limitations and Strengths

This study has several limitations. Firstly, this study population from multiple centers cannot be representative of all pregnant women in South Korea, although the population in most of the provinces was included. Secondly, responding women might have positive attitudes toward vaccination, which can result in selection bias. Thirdly, this survey did not include questions about income levels, to decrease dropout rates. Lastly, the number of non-respondents among the surveyed pregnant women was not recorded, although the response rate was estimated to be over 80% by the investigators, based on the number of pregnant or postpartum women who visited hospitals or clinics during the study period. This study was performed in primary, secondary, and tertiary hospitals to minimize bias, and the survey was conducted anonymously, allowing free expression of opinions by the respondents. 

Findings from this study add information about obstacles to and motivators for COVID-19 vaccination during pregnancy in South Korea. The vaccination efforts should specifically include providing safety and efficacy information about the COVID-19 vaccine. Several studies in pregnant women have demonstrated no evident differences between newborns of women who received BNT162b2 or mRNA-1273 mRNA vaccination during pregnancy and those of women who were not vaccinated [15,19,41]. These studies contribute to current evidence in establishing the safety of prenatal vaccine exposure to the newborns. In addition, vaccine-elicited antibodies were transported to infant cord blood and breast milk, and cross-reactive antibody responses and T-cell responses against SARS-CoV-2 variants were found in pregnant and nonpregnant women [21]. Other studies reported that milk from vaccinated lactating women exhibited neutralization activity against live SARS-CoV-2 virus and its variants, even 90 days after the second dose [42,43,44]. A recent study revealed a lower risk of a positive test for SARS-CoV-2 during the first 4 months of life among infants born to vaccinated mothers during pregnancy compared with infants of unvaccinated mothers [45]. It was reported that preterm delivery rates were from 14.3% to 28.1%, in women with COVID-19 infection [6,46], which is higher than the overall incidence (about 11%) of preterm delivery [47]. Vaccine hesitancy is defined by the World Health Organization as the delay in acceptance or refusal of vaccination despite availability [48]. At its core, vaccine hesitancy results from the decision-making process after considering a multitude of factors that influence that decision. A recent guideline of the Korean Society of Contraception and Reproductive Health also recommended vaccination for pregnant women [49]. Because top concerns about vaccine hesitancy are different in each country [50], understanding the specific concerns among individuals through social media can help to inform public communications. Although postvaccination surveillance, in particular after first trimester vaccination and long-term follow-up, in pregnant women and newborns, should be continued, updated information about effectiveness and safety should be transferred in a timely manner to pregnant women, especially by OBGYN doctors.

## 5. Conclusions

The acceptance rate of COVID-19 vaccination among pregnant women was less than 30% in Korea, which was much lower than those of influenza or Tdap. Because information from OBGYN doctors is critical to the acceptance of COVID-19 vaccination in pregnant women, OBGYN doctors should keep up with the new data about the disease, variants, severity in pregnant women, and vaccine efficacy and safety and provide those to pregnant women for their informed decision-making.

## Figures and Tables

**Figure 1 jcm-11-05733-f001:**
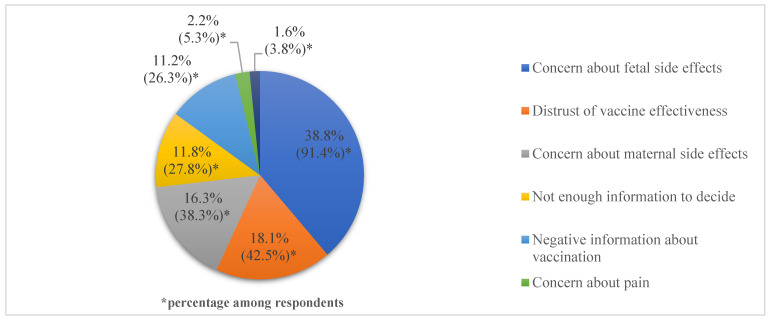
Concerns of pregnant or postpartum women about COVID-19 vaccination.

**Figure 2 jcm-11-05733-f002:**
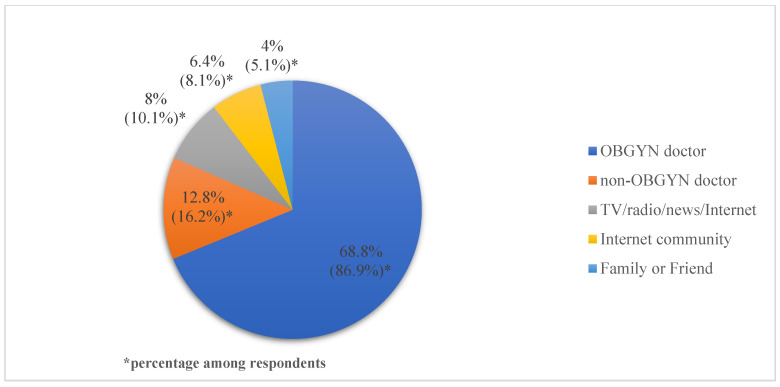
Information and recommendation routes that affect future COVID-19 vaccination. OBGYN, obstetrics and gynecology, TV, Television.

**Table 1 jcm-11-05733-t001:** Sociodemographic characteristics regarding COVID-19 vaccine acceptance.

Characteristics	Acceptance (*n* = 116)	Non-Acceptance (*n* = 320)	*p*-Value
Maternal age (years)	33.28 ± 4.70 ^a^	33.65 ± 3.77	0.445
Parity	42 (36.2)	140 (43.8)	0.158
Pre-pregnancy BMI (kg/m^2^) ^a^	22.87 ± 4.46 ^a^	22.42 ± 3.96	0.320
Natural conception	99 (85.3)	263 (82.2)	0.438
Capital region	79 (68.1)	199 (62.2)	0.256
Occupation			0.020 *
Housewife	47 (40.5)	135 (42.2)	
Non-medical personnel	51 (44.0)	163 (50.9)	
Medical personnel	18 (15.5)	22 (6.9)	
College degree or above	100 (86.2)	281 (87.8)	0.655
Comorbid disease	21 (18.1)	58 (18.1)	0.996
Pregnancy complications ^b^	16 (13.8)	35 (10.9)	0.412
Pregnancy period			0.001 **
1st trimester	2 (1.7)	29 (9.1)	
2nd trimester	29 (25)	47 (14.7)	
3rd trimester	78 (67.2)	198 (61.9)	
Postpartum	7 (6.0)	46 (14.4)	
Influenza vaccination during pregnancy	76 (65.5)	203 (63.4)	0.689

^a^ BMI, body mass index. ^b^ Pregnancy complications include gestational diabetes, pregnancy associated hypertension, and so on. Values are presented as the mean ± standard deviation or *n* (%). * *p* < 0.05, ** *p* < 0.01.

**Table 2 jcm-11-05733-t002:** COVID-19-related characteristics regarding COVID-19 vaccine acceptance.

Characteristics	Acceptance (*n* = 116)	Non-Acceptance (*n* = 320)	*p*-Value
Informed about COVID-19 vaccine	110 (94.8)	264 (82.5)	0.001 **
Informed by OBGYN ^a^ doctor	54 (46.6)	55 (17.2)	<0.001 ***
Informed by non-OBGYN doctor	8 (6.9)	11 (3.4)	0.118
Informed by public health office	12 (10.3)	15 (4.7)	0.030 *
Informed by media	77 (66.4)	224 (70)	0.470
Informed by family or friend	36 (31.0)	83 (25.9)	0.291
Cohabitant COVID-19 vaccination	108 (93.1)	266 (83.1)	0.008
Worried about COVID-19 infection during pregnancy	45 (38.8)	159 (49.7)	0.044 *
Worried about maternal risk of COVID-19 infection during pregnancy	31 (26.7)	96 (30)	0.506
Worried of fetus risk of COVID-19 infection during pregnancy	40 (34.5)	107 (33.4)	0.838
Aware of high-risk group among pregnant women when infected	111 (95.7)	294 (91.9)	0.171
Aware that antibodies from vaccination can transfer to fetus	92 (79.3)	218 (68.1)	0.023 *

^a^ OBGYN, obstetrics and gynecology. Values are presented as *n* (%). * *p* < 0.05, ** *p* < 0.01, *** *p* < 0.001.

**Table 3 jcm-11-05733-t003:** Associated factors for maternal COVID-19 vaccination acceptance.

Factors	Univariate			Multivariate		
	COR ^a^	95% CI ^b^	*p*-Value	AOR ^c^	95% CI	*p*-Value
Occupation						
Housewife	1			1		
Non-medical personnel	0.90	0.57–1.42	0.647	0.85	0.52–1.40	0.523
Medical personnel	2.35	1.16–4.76	0.018	1.99	0.92–4.32	0.081
Pregnancy period						
1st trimester	1			1		
2nd trimester	8.95	1.99–40.33	0.004	7.35	1.54–35.15	0.012
3rd trimester	5.71	1.33–24.51	0.019	3.85	0.84–17.55	0.082
Postpartum	2.21	0.43–11.36	0.344	2.13	0.38–11.82	0.388
Informed about COVID-19 vaccine	3.89	1.63–9.29	0.002	2.08	0.83–5.23	0.118
Informed by OBGYN ^d^ doctor	4.20	2.63–6.69	<0.001	3.41	2.05–5.65	<0.001
Informed by public health office	2.35	1.06–5.18	0.035	1.51	0.64–3.58	0.351
Cohabitant COVID-19 Vaccination	2.74	1.26–5.95	0.011	2.43	1.06–5.59	0.036
Worried about COVID-19 infection during pregnancy	0.64	0.42–0.99	0.045	0.74	0.46–1.19	0.213
Aware that antibodies from vaccine can transfer to fetus	1.79	1.08–2.98	0.024	1.43	0.83–2.49	0.201

^a^ COR, crude odds ratio; ^b^ CI, confidence level. ^c^ AOR, adjusted odds ratio, adjusted for maternal age, occupation, and pregnancy period. ^d^ OBGYN, obstetrics and gynecology.

## Data Availability

The data presented in this study are available upon request from the corresponding author.

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
