# Peer review of "COVID-19 Vaccine Acceptance during Pregnancy and Influencing Factors in South Korea"

_jcm, 2022, doi:10.3390/jcm11195733_

Round 1

Reviewer 1 Report

1) The introduction is very good, but it will be nice if the authors mentioned which vaccines has been used in Korea to vaccinate pregnant woman. Also since when the pregnant woman were vaccinated in Korea. 

2) Add to the manuscript if the protocol was approved by any ethical committee. Without this information, is not ethical to publish de article or performed a research. 

3) Which test was usted to determined the data distribution or normality. 

4) In Table 1 the occupation and pregnancy period were statistically significative, the authors did correct the OR with those factors? If not, please correct the OR of table 3 not only by maternal age, also by occupation and pregnancy period. 

5) In discussion compare deeper their results with other countries and with other vaccines. 

Author Response

Thank you for your valuable comments. We revised it according to the comments.

1) The introduction is very good, but it will be nice if the authors mentioned which vaccines has been used in Korea to vaccinate pregnant woman. Also since when the pregnant woman were vaccinated in Korea. 

   We added which vaccines were provided in pregnant women and the timepoint, in the introduction section, based on the comment.

2) Add to the manuscript if the protocol was approved by any ethical committee. Without this information, is not ethical to publish de article or performed a research. 

According to the comment, we wrote it in the method section as follows; Ethical approval for protocol and questionnaires was granted by the Institutional Review Boards of the Catholic University of Korea (KC21QIDE0934) and Seoul National University (2203-101-1308).

3) Which test was used to determined the data distribution or normality. 

Data distribution was determined by F test. We wrote in the method section.

4) In Table 1 the occupation and pregnancy period were statistically significative, the authors did correct the OR with those factors? If not, please correct the OR of table 3 not only by maternal age, also by occupation and pregnancy period. 

Data were adjusted for maternal age, occupation, and pregnancy period. We added it, in the footnote of table 3 and method section.

5) In discussion compare deeper their results with other countries and with other vaccines. 

We added more comparisons in the discussion section, as follows.

Other predictors in a survey from 16 countries were worrying about COVID-19, belief in the importance of vaccines to their own country, mask-wearing compliance, trust of public health agencies, and attitudes towards routine vaccines [35]. In this study, women in non-acceptance group more worried about COVID-19, compared to women in acceptance group, in univariate analysis. However, it was not significant in multivariate analysis. In addition, there were no significant differences whether they were informed from public health office, media, non-OBGYN doctors, and family members or friends, in this study. Because 69% of women responded that information and recommendation from OBGYN doctors would affect future COVID-19 vaccination, providing information and guidelines from academic committees and public health office to OBGYN doctors, it seems more important to increase vaccine acceptance in South Korea, compared to in other countries.  

Reviewer 2 Report

Estimated Authors,

Estimated Editors,

I've read with great interest the present paper from the study group led by Yoon Heesu, and reporting on the acceptance of COVID-19 vaccination in pregnant women. More precisely, Authors identified factors such as having received "vaccine information about from obstetrics and gynecology (OBGYN) doctors” (OR 3.41, 95% CI 2.05–5.65), living with individuals having received COVID-19 vaccination (OR 2.43, 95% CI 1.06-5.59), and being in the second trimester of pregnancy as main predictors (OR 7.35, 95% CI 1.54-35.15). Such estimates are reinforced by a preventive sample size calculation, and accurately discussed by Authors in discussion and conclusion sections.

In fact, the present Author recommends the acceptance of this study as it is, being the present article of very good quality.

Author Response

Thank you for your comment.

We clearly described main findings as you suggested, in the discussion, as followed.

However, we identified factors such as having received "vaccine information about from OBGYN doctors” (OR 3.41, 95% CI 2.05–5.65), living with individuals having received COVID-19 vaccination (OR 2.43, 95% CI 1.06-5.59), and being in the second trimester of pregnancy as main predictors (OR 7.35, 95% CI 1.54-35.15), in multivariate logistic regression analysis.

Reviewer 3 Report

Although the topic is of public health interest, the study methods were rather problematic and the manuscript also requires close edits for language and style.

Specific comments:

1. Please spell abbreviations such as "CDC" and "VAERS" out in full in the first instance of their use.

2. "Coronavirus 2019 (COVID-19)" - Please note that "COVID-19" is actually short for "Coronavirus Disease 2019".

3. Please change "Though such vaccine hesitancy during pregnancy is not so new" to "Though such vaccine hesitancy during pregnancy is not new".

4. Vaccine hesitancy is defined by the World Health Organisation (WHO) as the delay in acceptance or refusal of vaccination despite availability. At its core, vaccine hesitancy results from the decision-making process after considering a multitude of factors which influence that decision. This is an important background that should be included in the introduction section.

5. Please change "Jan 01, 2022, to Apr 30, 2022" to "1 Jan 2022 to 30 Apr 2022".

6. "... a mix of public and private clinics or hospitals located in South Korea" - which cities in South Korea, and how many clinics and hospitals exactly? Is it only Seoul since the IRB approval comes from two Universities in Seoul? More details are necessary.

7. Are the authors able to estimate the response rate for the study? This is important as a low response rate can contribute to non-response bias or a negative bias among respondents.

8. Please calculate and present the survey completion rate.

9. "320 pregnant women in non-acceptance group were" - do not begin the sentence with a number.

10. In the discussion section, suggest mentioning some of the prevailing negative sentiments regarding COVID-19 vaccination based on social media analyses, i.e. emotional reactions to perceived invidious policies or safety and effectiveness concerns related to the COVID-19 vaccines (citation: doi 10.3390/vaccines10091457). This is relevant to the patient population under study and would affect one's willingness to get vaccinated.

11. Figures 1 and 2 should be coloured to allow easy visualization.

12. "This study indicated a lower rate of acceptance of COVID-19 vaccination among pregnant women than in the general population" - where or how is this figure of reference derived? Is this based on historical surveys or a control group? This is important to mention as you may not be comparing apple for apple; confidence in COVID-19 vaccines has probably waned compared to early in the pandemic given a significant proportion of COVID recovered individuals and growing concerns regarding the lack of effectiveness of ancestral COVID-19 vaccines against new VOCs.

13. How does one get informed by the "public health office"? More information is necessary. Is this a routine phone call, email, or text message sent to pregnant women or vulnerable persons in Korea?

14. Please change "We thank pregnant and postpartum women" to "We thank all pregnant and postpartum women".

Author Response

Thank you for your valuable comments. We revised the manuscript, based on the comment.

1. Please spell abbreviations such as "CDC" and "VAERS" out in full in the first instance of their use.  : We described full name in the manuscript.

2. "Coronavirus 2019 (COVID-19)" - Please note that "COVID-19" is actually short for "Coronavirus Disease 2019".: We described full name in the manuscript.

3. Please change "Though such vaccine hesitancy during pregnancy is not so new" to "Though such vaccine hesitancy during pregnancy is not new". : We changed it, according to the comment.

4. Vaccine hesitancy is defined by the World Health Organisation (WHO) as the delay in acceptance or refusal of vaccination despite availability. At its core, vaccine hesitancy results from the decision-making process after considering a multitude of factors which influence that decision. This is an important background that should be included in the introduction section.

: Because introduction already included too much contents, we included the content about vaccine hesitancy, in the discussion section.

5. Please change "Jan 01, 2022, to Apr 30, 2022" to "1 Jan 2022 to 30 Apr 2022".: We changed it.

6. "... a mix of public and private clinics or hospitals located in South Korea" - which cities in South Korea, and how many clinics and hospitals exactly? Is it only Seoul since the IRB approval comes from two Universities in Seoul? More details are necessary.

: We wrote details in the method section, as followed.

 Four hospitals in Seoul, one hospital in Daegu, one hospital in Kangwon, and one clinic in Jeolla were included. 

7. Are the authors able to estimate the response rate for the study? This is important as a low response rate can contribute to non-response bias or a negative bias among respondents.

: We did not record response rates. Based on the answers from investigators, we wrote it in the discussion section, as followed.

The number of non-respondents among the surveyed pregnant women was not recorded, although response rate was estimated over 80% by investigators.

8. Please calculate and present the survey completion rate.

: We wrote it in the result section, based on the comment.

-> Survey completion rate was 93.8% (436/465).

9. "320 pregnant women in non-acceptance group were" - do not begin the sentence with a number.

: We changed it, based on the comment.

Women in non-acceptance group (n=320) were asked why they did not receive vaccination, and multiple responses were allowed.

10. In the discussion section, suggest mentioning some of the prevailing negative sentiments regarding COVID-19 vaccination based on social media analyses, i.e. emotional reactions to perceived invidious policies or safety and effectiveness concerns related to the COVID-19 vaccines (citation: doi 10.3390/vaccines10091457). This is relevant to the patient population under study and would affect one's willingness to get vaccinated.

: Because there is no citation as doi 10.3390/vaccines10091457, we added a similar article and mentioned it, in the discussion section.

Because top concerns about vaccine hesitancy were different in each country [51], understanding the specific concerns among individuals through social medias can help to inform public communications. 

11. Figures 1 and 2 should be coloured to allow easy visualization.

: We changed those as colored figures.

12. "This study indicated a lower rate of acceptance of COVID-19 vaccination among pregnant women than in the general population" - where or how is this figure of reference derived? Is this based on historical surveys or a control group? This is important to mention as you may not be comparing apple for apple; confidence in COVID-19 vaccines has probably waned compared to early in the pandemic given a significant proportion of COVID recovered individuals and growing concerns regarding the lack of effectiveness of ancestral COVID-19 vaccines against new VOCs.

: We added percentage and reference in the discussion section. This study indicated a lower rate of acceptance of COVID-19 vaccination among pregnant women than in the general population, which was reported as 80% in South Korea, until 29 Oct 2021 [29].

13. How does one get informed by the "public health office"? More information is necessary. Is this a routine phone call, email, or text message sent to pregnant women or vulnerable persons in Korea?

: We added it in the method section, according to the comment. General information about COVID-19 vaccination for pregnant women was sent by text message and mailing from regional public health offices, when vaccination program for pregnant women commenced.

14. Please change "We thank pregnant and postpartum women" to "We thank all pregnant and postpartum women". : We changed it.

Round 2

Reviewer 3 Report

Thank you for the revisions.

Specific comments:

1. Please change "... was expected 20%" to "... was expected to be around 20%". Please also provide a citation to support this assumption.

2. "Because there is no citation as doi 10.3390/vaccines10091457, we added a similar article and mentioned it, in the discussion section" - the citation for this is: Ng QX, Lim SR, Yau CE, Liew TM. Examining the Prevailing Negative Sentiments Related to COVID-19 Vaccination: Unsupervised Deep Learning of Twitter Posts over a 16 Month Period. Vaccines. 2022; 10(9):1457. Please correct accordingly.

3. Please change "were expressed as number (%)" to "were expressed as counts and percentages".

4. "... estimated over 80% by investigators" - how was this estimated? This was unclear to readers.

5. "... and possibly incomplete" - this can be omitted.

Author Response

Thank you for your valuable comments.

We revised our manuscript, based on the comments. Corrected parts were highlighted with yellow color.

  1.  Please change "... was expected 20%" to "... was expected to be around 20%". Please also provide a citation to support this assumption. : We changed it and added a citation.

2. "Because there is no citation as doi 10.3390/vaccines10091457, we added a similar article and mentioned it, in the discussion section" - the citation for this is: Ng QX, Lim SR, Yau CE, Liew TM. Examining the Prevailing Negative Sentiments Related to COVID-19 Vaccination: Unsupervised Deep Learning of Twitter Posts over a 16 Month Period. Vaccines. 2022; 10(9):1457. Please correct accordingly.  : We changed the reference, based on the comment.

3. Please change "were expressed as number (%)" to "were expressed as counts and percentages".  : We changed it.

4. "... estimated over 80% by investigators" - how was this estimated? This was unclear to readers.   : We added it.

5. "... and possibly incomplete" - this can be omitted.: We removed it.